# A Bibliometric Analysis on the Research Trend of Exercise and the Gut Microbiome

**DOI:** 10.3390/microorganisms11040903

**Published:** 2023-03-30

**Authors:** Ruiyi Deng, Mopei Wang, Yahan Song, Yanyan Shi

**Affiliations:** 1Research Center of Clinical Epidemiology, Peking University Third Hospital, Beijing 100191, China; 2Department of Medical Oncology and Radiation Sickness, Peking University Third Hospital, Beijing 100191, China; 3Library, Peking University Third Hospital, Beijing 100191, China

**Keywords:** physical activity, gut microbiota, disease, metabolic syndrome, cancer, probiotics

## Abstract

This article aims to provide an overview of research hotspots and trends in exercise and the gut microbiome, a field which has recently gained increasing attention. The relevant publications on exercise and the gut microbiome were identified from the Web of Science Core Collection database. The publication types were limited to articles and reviews. VOSviewer 1.6.18 (Centre for Science and Technology Studies, Leiden University, Leiden, the Netherlands) and the R package “bibliometrix” (R Foundation: Vienna, Austria) were used to conduct a bibliometric analysis. A total of 327 eligible publications were eventually identified, including 245 original articles and 82 reviews. A time trend analysis showed that the number of publications rapidly increased after 2014. The leading countries/regions in this field were the USA, China, and Europe. Most of the active institutions were from Europe and the USA. Keyword analysis showed that the relationship between disease, the gut microbiome, and exercise occurs throughout the development of this field of research. The interactions between the gut microbiota, exercise, status of the host’s internal environment, and probiotics, are important facets as well. The research topic evolution presents a trend of multidisciplinary and multi-perspective comprehensive analysis. Exercise might become an effective intervention for disease treatment by regulating the gut microbiome. The innovation of exercise-centered lifestyle intervention therapy may become a significant trend in the future.

## 1. Introduction

The human gastrointestinal (GI) tract harbors a microbial diversity of up to 100 trillion microorganisms, including bacteria, archaea, viruses, and parasites, commonly referred to as the gut microbiome [1,2,3]. Having coevolved with the host for thousands of years, the gut microbiome has formed an intricate and mutually beneficial relationship [4,5]. It has been proven that the gut microbiome plays a significant role in the maintenance of physiological homeostasis [5,6,7,8]. The gut microbiome constitutes a changing ecosystem influenced by numerous host-dependent factors, such as genetic background, age, sex, diet, lifestyle, stress, antibiotic use, diet, and socioeconomic aspects [9,10,11,12]. In recent years, there has been growing interest in the relationship between lifestyle and the gut microbiome [10]. An active lifestyle, including regular physical exercise, is recommended to prevent disease and preserve health, in all stages of life [10,13,14]. Exercise is considered an optimal method to prevent several diseases of the cardiovascular, neuroendocrine, respiratory, and muscular systems and an efficient nonpharmacological way to lower the risk of metabolic syndromes, various cancers, and some mental illnesses [13,14,15]. Although studies have shown that these diseases are associated with the gut microbiome, the relationship between exercise, the gut microbiome, and disease has not been classified. Exercise can modify the diversity, composition, and functionality of gut microbial populations and increase biodiversity and the presence of taxa with beneficial metabolic functions [16,17,18], which may be associated with some metabolic syndromes, cardiovascular diseases, cancers, and neuropsychiatric diseases [19,20,21,22,23]. In addition, modifications of the gut microbiome composition may support training, performance, and post-exercise recovery of the hosts [18,24,25,26]. In-depth study of the gut microbiome and exercise will help to improve the understanding of the microbiome’s contribution to host health and disease, further optimize the method of exercise intervention, expand its application in disease prevention and treatment, and develop potential therapies to improve sports performance by regulating the gut microbiome.

Comprehensively assessing the status of research and scientific output on the gut microbiome and exercise, is beneficial for researchers to reasonably determine future research directions and improve research efficiency. Therefore, this study aims to provide an overview of research trends on the microbiome and exercise at the global level, and to visualize the research hotspots and trends through a bibliometric analysis. Different from systematic reviews and scoping reviews, bibliometric analysis is based on the unique parameters of publications, which uses a combination of mathematics and statistical methods to quantitatively describe the current status and research hotspots of a field’s national and worldwide contributions to the literature trends in science and technology [27,28]. By visualizing the research trends, bibliometric analysis may help researchers to achieve a special purpose within their domains of interest (for example, identify active partners, landmark studies, research themes, and other useful information) and identify research gaps that might be addressed in future studies [27,28,29,30]. To comprehensively capture the current status and trends in gut microbiome and exercise research, this study provides an objective assessment of the contributions of academic groups and individual researchers and reveals current research trends and hotspots, which will provide a basis for future research.

## 2. Materials and Methods

The Web of Science Core Collection (WOSCC) is an optimal database, with more than 20,000 high-quality journals, and is the most commonly used database in bibliometric analyses [31]. All relevant studies for this bibliometric analysis, were identified from the WOSCC database on 19 October 2022. The study strategy was the combination of exercise and gut microbiota using the following words: TS = (‘Exercise’ OR ‘Physical Activity’ OR ‘Sports’) and TS = (‘Gut Microbiome’ OR ‘Gut Microbiota’ OR ‘Intestinal Microflora’). The detailed search strategy is provided in the Appendix A.

Two authors (Ruiyi Deng and Mopei Wang) independently evaluated the titles and abstracts of all relevant publications, and any differences were resolved by full discussion; otherwise, conflicts were resolved by a third author (Yanyan Shi). Microsoft Office Excel 2019 (Microsoft, Redmond, WA, USA) was used to collect the extracted data and analyze the data from the WOSCC. The R package “bibliometrix” (version 3.2.1) (https://www.bibliometrix.org, accessed on 7 February 2023) was applied, to quantify parameters such as the number of publications, countries/regions, authors, institutions, journals, and citations, and to visualize the results [32]. The countries/regions collaboration map and conceptual structure maps generated by the software package, can reflect the research trends of a certain discipline or knowledge field in a certain period, facilitating an accurate understanding of the evolution of a given scientific frontier. VOSviewer 1.6.18 (Centre for Science and Technology Studies, Leiden University, Leiden, the Netherlands) was used to access and graph the co-authorship, co-occurrence, and times cited of countries/regions, institutions, authors, hotspot keywords, and journals. Different colors represent the different modules in the visualization mapping. Each node in the figure represents a country/region or institution, and these nodes were clustered into different groups (colors) according to their cooperation. The size of the node represents the occurrence frequency of the relevant parameters, and the thickness of the connection between nodes represents the degree of cooperation.

## 3. Results

### 3.1. General Description of Study Selection and Characteristics

As shown in Figure 1, a total of 4284 publications were identified from the WOSCC database, and 176 of them were excluded because they were not published in original articles or reviews. The titles and abstracts of the remaining 4108 publications were carefully screened by the two investigators, and 327 relevant publications were eventually included. There were 245 original articles, accounting for 75% of the total records, and the other 82 publications were reviews (Figure 2A).

The number of publications in each period reflects the development trend of research in this field. According to the global annual number of publications in the fields of the gut microbiome and exercise (Figure 2B), the growth in the publications can be divided into two periods: 2005–2013 and 2014–2022. Before 2014, there were few studies in this field each year, and publications during this period accounted for only 2.1% (*n* = 7) of the total. The number of publications rapidly increased after 2014, and the peak of annual publications seems to be yet to come, reaching 70 in 2021.

As seen from the average annual citations (Figure 2C), the citation frequency was low from 2000 to 2005, indicating that this research was still in its infancy and did not receive widespread attention. During 2005–2014, with the increasing annual number of publications in the fields of gut microbiome and exercise, the annual total citations showed a fluctuating increase, and the highest yield of cited papers was in 2014, at 25.67. In 2014–2022, although the volume of published papers showed a steady upward trend, the average citation of papers declined. This indicates that, despite the various research directions in the fields of the gut microbiome and exercise, the influence of the papers was decreasing.

### 3.2. Analysis of Active Countries/Regions, Institutions and Authors

The global distribution and international cooperation network of countries/regions is shown in Table 1 and Figure 3. Among the 10 most productive countries/regions, the USA had the largest number of studies (81 records), followed by China (64 records), Italy (38 records), Spain (26 records), and England (24 records). The USA and China contributed 33.48% of the total publications, far more than any other country/region (Figure 3A). Figure 3B presents the annual distribution of research from the top 10 countries/regions. The annual number of publications from China continues to grow rapidly, but the annual proportion of research from China has fluctuated in recent years (Figure 3C). The USA had by far the most total citations of all countries/regions (3218 times cited), followed by China (1254 times cited), Ireland (1215 times cited), Spain (1166 times cited), and Italy (664 times cited). In terms of average article citations, Monaco showed the highest average number of citations (131.00), followed by Ireland (93.46), Spain (77.73), the USA (53.63), and Canada (41.90) (Figure 3D). Although China had more total citations, the average citations were significantly lower than those of other countries/regions, because of the large number of papers and relatively low quality of many of these papers. The cooperation cluster network of countries/regions connected to each other is shown in Figure 3A,E. Four clusters were obtained based on this information. Twenty-nine countries/regions published more than three publications, and 27 of them connected to each other to form the cooperation network. The USA was the most cooperative country/region (total link strength (TLS) = 50). A total of 18 countries/regions had cooperative relations with the USA. China, Canada, and England cooperated most closely with the USA. China cooperated with 10 countries/regions and cooperated closely with the USA, Japan, and Taiwan. Thus, considering the quantity, quality, and impact of the publications and international cooperation, the USA revealed a leading position in the field of the microbiome and exercise.

The cooperation cluster network of institutions connected to each other is shown in Figure 4A. A total of 83 institutions are displayed. According to their cooperation, the 83 institutions are clustered into 13 groups (colors). As shown in Table 2, the University of Illinois, from the USA, was the most productive institution (11 records) and had the highest number of times cited (1043), followed by Paris-Saclay University, from France (9 publications with 556 times cited), Sapienza University of Rome, from Italy (7 publications with 138 times cited), and University College Cork, from Ireland (7 publications with 127 times cited). The University of Illinois cooperated closely with Sapienza University of Rome, from Italy, the University of Missouri, from USA, and Xi’an Jiaotong University, from China. Zonguldak Bulent Ecevit University, from Turkey, had the highest number of times cited per publication (210). Sapienza University of Rome had the most co-operations (TLS = 17), which cooperated closely with the University of Physical Education, from Hungary, Foro Italico University of Rome, from Italy, Pazmany Peter Catholic University, from Hungary, Waseda University, from Japan, and the University of Illinois, from the USA. There are 16 universities from China that published more than two publications, 11 of which were included in the cooperation cluster network. Among them, Shanghai University of Sport was the most productive and the most cooperative institution (4 publications with 82 times cited). The main cooperating institutions of Shanghai University of Sport include Appalachian State University, Shanghai Jiao Tong University, Guangzhou Sport University, and Shandong Sport University.

A total of 2255 authors published papers on this topic, of which Cotter, P.D., Huang, W.C., and Mach, N. were the top three authors, with 7, 6, and 6 publications, respectively (Table 3). In this field, Prof. Cotter, P.D. had the largest number of publications, with an h-index of 5, g-index of 7, and total number of citations of 1101. This indicates the high quality and great influence of Cotter’s papers. As shown in Figure 4B, Cotter’s most frequently cited paper appeared in 2014 (the darkest color of the graph), which was cited 675 times. Cotter’s article entitled “Exercise and associated dietary extremes impact on gut microbial diversity”, published in *Gut*, demonstrated that the relationship between exercise and gut microbiota is complex and is related to accompanying dietary extremes, providing evidence for a beneficial impact of exercise on gut microbiota diversity [33]. Combining productivity and impact, the h-index has been considered a widely accepted tool for measuring scientific performance, assessing current paper volumes, and predicting authors’ future performance [34,35,36]. Mach, N. had the highest h-index in this field. Mach, N. has been publishing papers in this field since 2016, while the most frequently cited year was 2017. For example, Mach, N.’s review entitled “The Crosstalk between the Gut Microbiota and Mitochondria during Exercise”, published in *Frontiers in Physiology*, summarized the advances in the interaction between mitochondria and gut microbiota during exercise, demonstrating that targeting the gut microbiota can be useful for managing mitochondrial-related ROS production, pro-inflammatory signals, and metabolic limits in endurance athletes [37]. In addition, the cooperation cluster network of authors is shown in Figure 4C,D. Among the 2255 authors, 189 have published more than 2 papers. However, the degree of cooperation among authors was relatively low, and only 22 of them were connected to each other, mainly consisting of authors from the University of Illinois, Mayo Clinic, and the University of Missouri.

### 3.3. Analysis of Relevant Journals and Highly Influential Publications

All of the included articles were published in 167 different journals. The most relevant journals were *Nutrients*, *PLoS ONE*, and *Frontiers in Microbiology,* with a total number of publications of 29, 12 and 9, respectively (Table 4). The top five journals published 64 (19.6%) papers. As shown in Figure 5A, the annual number of the top five journals’ publications has gradually increased since 2014. The journal *Nutrients* had the highest growth rate of the annual number of published papers, whereas *Gut* (928) and *PLoS ONE* (829) had the highest total local citations. Data from the 2021 edition of journal citation reports, showed that among the top 10 journals, the journal with the highest impact factor is *Gut Microbes* (IF = 9.434), followed by *Msystems* (IF = 7.328). For the h-Index, the journals with the greatest impact were *Nutrients* (h-index = 13), *PLoS ONE* (h-index = 10), *Gut Microbes* (h-index = 6), and *Frontiers in Microbiology* (h-index = 6). According to Bradford’s Law, the source journals of research papers in the field of exercise and the microbiome were highly scattered, and 14 of the 167 journals were selected as core sources, including the top 10 journals mentioned above (Figure 5B). These journals played an essential role in the field of exercise and the microbiome during the study period.

The details of the ten most cited studies in the WOSCC database are shown in Table 5. The top ten highest times cited ranged from 187 to 644; two studies were published in the *Gut* journal, and the remaining eight studies were published in different journals. The ten studies consist of eight articles and two reviews [15,17,33,38,39,40,41,42,43,44]. The most cited studies explore the impact of exercise and associated dietary extremes on gut microbial diversity [33]. Three studies explored the interplay between exercise and the gut microbiome in obese humans [39,41,42]. Three studies investigated the impact of exercise on the gut microbiome [15,33,38]. Two other studies investigated the gut microbiome among athletes [41,42].

### 3.4. Analysis of Keywords

Keywords represent the high generalization of a research topic and content. Analysis of high-frequency keywords could reflect the research trend and hotspots in the field of exercise and the microbiome in a straightforward way. A cloud map of the top 100 keywords is shown in Figure 6A, with larger fonts indicating a higher frequency of occurrence. High-frequency keywords included “exercise”, “health”, “diet”, “inflammation”, “intestinal microbiota”, “obesity”, “physical-activity”, “gut microbiota”, “chain fatty-acids”, and “diversity”, all occurring more than 30 times. Figure 6B describes the evolution trend of core keywords over time. The positions of the green, red, and blue dots represent the first quantile, the third quantile, and the median of the publication year corresponding to the keyword, respectively. The position of the red dots and the size of the blue dots can reflect the research trends. The farther the red dot is to the right of Figure 6B and the larger the blue dot, the more recent the publication and the more frequent occurrence in papers published for the corresponding keyword, respectively. It was observed that the KeyWords Plus used in 2020 were the most often repeated, including “health”, “exercise”, and “diet”, which indicates that the research in this period mainly focused on the interaction between lifestyle and health. Then, in 2021, the main keywords changed to “metabolism”, “chain-fatty acids”, and “association”, reflecting that the relationship between exercise, metabolism, and the gut microbiome gained attention in this period. The temporal trend analysis showed that keywords including “obese”, “host”, “cells”, “metabolism”, “chain-fatty acids”, “association”, and “exercise” represent the most recent hotspots. The change in metabolism during exercise and the role of exercise in ameliorating metabolic syndromes may be one of the current research hotspots. Moreover, the research topic and anticipated research direction evolve over time. The thematic evolution detailed the evolutionary associations that displayed field development, presented quantitative evidence such as thematic flow, direction of thematic flow, and conversion associations, and showed the point of development direction and evolution routes [45,46]. A Sankey diagram was used to interpret the thematic evolution in the field of exercise and the gut microbiome from 2000 to 2022 (Figure 6C), and to visualize and categorize the detected themes of the studied field [46]. The size of the nodes in the Sankey diagram is commensurate with the number of keywords included in the subject. The keywords in different periods are connected by gray lines, representing the research topic’s evolving focus. The thicker the line, the greater the significance of the two themes over the year. The period considered for our collection of publications was split into four smaller periods: 2000–2013, 2014–2017, 2018–2020, and 2021–2022. In the first period, 2000–2013, research in the field of exercise and the gut microbiome was in its infancy, and scientists began to investigate subjects such as “exercise”, “microflora”, and “obesity”. From 2014 to 2017, studies on exercise and the gut microbiome gradually increased, mainly focusing on mechanisms and pathological processes associated with the gut microbiome, such as “intestinal microbiota”, “immunity”, “inflammation”, “receptor”, “community”, “chronic psychological stress”, “responses”, “responses”, and “bacterial translocation”. From 2018 to 2020, the field of study gradually expanded to the relationship between diseases, gut microbiome, and exercise, which includes “exercise”, “gut microbiome”, “immunity”, “inflammatory-bowel disease”, “inflammation”, “oxidative stress”, “responses”, “insulin-resistance”, “overweight”, “insulin sensitivity”, and “cardiorespiratory”. From 2021 to 2022, the results demonstrate that research topics evolved as an emerging theme, while the relationship between disease, the gut microbiome, and exercise remains the primary theme, and some new thematic areas emerged: “weight-loss”, “aerobic exercise”, “skeletal-muscle”, “alters”, “dysbiosis”, and “blood”. In short, the number of publications on exercise and the microbiome has rapidly increased over time, and the research topic evolution has been ongoing, presenting a trend of multidisciplinary and multi-perspective comprehensive analyses.

We manually standardized the keywords, before constructing the cluster map. We merged and replaced similar keywords. Terms with a minimum number of occurrences greater than five, in all included publications, were analyzed using VOSviewer 1.6.18 (Centre for Science and Technology Studies, Leiden University, Leiden, the Netherlands). There were 135 terms that reached this threshold, out of 1486 in this field, which were divided into six clusters and colored differently (Figure 7A). The three most common keywords were “gut microbiota” (TLS: 1469, occurrences: 259), “exercise” (TLS: 1143, occurrences: 197), and “obesity” (TLS: 498, occurrences: 78). Cluster 1 comprised 29 items, that represented the interactions among the gut microbiota, exercise, and physiological state of the host (shown in red), including keywords such as “gut microbiota”, “exercise”, “aerobic exercise”, “gut health”, “brain”, “depression”, “oxidative stress”, “fitness”, “cardiorespiratory fitness”, “intestinal permeability”, “anxiety”, and “cognitive functions”. Cluster 2 comprised 29 items, that mainly represented probiotics and exercise performance (shown in green), including keywords such as “probiotics”, “supplementation”, “dietary fiber”, “fermentation”, “athletes”, “endurance exercise”, “muscle”, and “athletic performance”. Cluster 3 comprised 25 items, that mainly contained keywords about the crosstalk between exercise and inflammation-related pathological processes in the gut, such as “swimming training”, “treadmill exercise”, “moderate exercise”, “inflammation”, “TNF-α”, “inflammatory-bowel-disease”, “intestinal barrier”, “receptor”, and “pathways” (shown in blue). Cluster 4 comprised 22 items, that represented dysfunction and disorders of the host’s internal environment, such as “dysbiosis”, “obesity”, “blood-pressure”, “insulin sensitivity”, “hypertension”, and “dysfunction” (shown in yellow). There were fifteen items in cluster 5 (shown in purple), which mainly concerned lifestyle and metabolic-related diseases, including “lifestyle”, “Mediterranean diet”, “consumption”, “overweight”, “metabolic syndrome”, “insulin resistance”, and “nonalcoholic fatty liver”. There were fifteen items in cluster 6 (shown in light blue), mainly consisting of keywords about general characteristics of human objects such as “gut”, “butyrate-producing bacterium”, “diversity”, “ecology”, “metabolism”, “aging”, “older adults”, and “populations”. Figure 7B shows the research trend. Currently, the relationship between diet, exercise, and gut microbiota is gaining increasing attention. As shown in the item and cluster density visualization of keyword co-occurrence (Figure 7C,D), research on the interaction between gut microbiota and exercise focuses on metabolism, obesity, metabolic syndrome, athletes, and probiotics.

## 4. Discussion

The current bibliometric study comprised comprehensive quantitative and qualitative analyses of the scientific output related to exercise and the gut microbiome, investigating the characteristics of countries/regions, journals, authors, and institutions contributing to a research area, revealing the features of keywords and the evolution of research subjects, and establishing cooperation networks between countries/regions, institutions, and authors. Our study not only presents current research trends and hotspots but also provides scholars with essential references and suggestions for further investigation on exercise and the gut microbiome.

### 4.1. General Information

In this study, the research trend analysis showed that annual publications before 2014 accounted for only 2.1% of the total, indicating that research on exercise and the gut microbiome was still in its infancy. Although the number of annual publications was low, the annual mean total citations showed a fluctuating increase, and the peak of cited papers occurred in 2014. The results demonstrate that studies in this period gradually brought attention to the field of exercise and the gut microbiome and became the basis for subsequent studies. The annual number of publications has continued to increase since 2014 and has not reached its peak. With an average annual growth rate of 42.9% and a steeper increase between 2014 and 2022 compared to the previous decade, research on exercise and the gut microbiome is in an explosive period, and related research has attracted increasing attention from scholars. With the increasing number of publications and the expansion of the research field, the annual mean total citations indicate that the average quality and impact of studies is declining.

According to our analysis, the geographical distribution covers all continents, and the predominant hubs for production are the USA, China, and Europe. The USA published the most studies and was the most influential country/region in this field (81 publications, 3218 times cited, h-index = 30). China had the highest number of publications and academic influence in Asia (64 publications, 1254 times cited, h-index = 18). The number of annual publications from China showed the fastest growth among the top 10 countries/regions, indicating that this field has recently gained increasing attention in China. China has gradually played an increasingly important role in the field of exercise and the gut microbiome, with the average annual proportion of the research being approximately 22%. However, the average citations of Chinese studies were significantly lower than those of other countries/regions, so the quality and impact of the studies need to be further improved. Academic capability mainly depends on the economic status of a country and its governmental expenditure on healthcare [47]. The outstanding economic status and large health expenditure of America may partly explain its high scientific production and excellent academic impact [47,48]. Regarding international collaboration, the USA was the most cooperative country/region (TLS = 50). China, Canada, and England were its main partners. Most of the international collaborations were between North America and European countries/regions. International collaborations with China were relatively low among the top 10 most productive countries/regions (TLS = 20). A greater collaboration network should be constructed in the future.

In regard to research institutions, most of the active institutions with high production and excellent academic impact were from Europe and America. There were 16 institutions from China publishing more than two publications, the distribution of which was relatively scattered. The cooperation among the institutions is mainly domestic cooperation, and international collaboration is still lacking. International collaboration among institutions is beneficial for leveraging the research strengths of the centers and improving the quantity and quality of publications. Therefore, collaborations among institutions need to be further enhanced in the future.

From the perspective of the authors, the top 10 authors remained stable over a long period. Prof. Cotter, P.D. was the most productive author, most of the studies of whom focus on the unique characteristics of the gut microbiome of athletes [49,50,51], compare the differences between elite athletes and sedentary subjects [41], and explore the effect of fitness improvement on the gut microbiome [33,52,53]. Prof. Cotter, P. D.’s work showed that the impact of exercise on the gut microbiome at the compositional, metagenomics, and metabolomic levels provides additional insights into the diet–exercise–gut microbiota paradigm, laying the foundation for subsequent studies. Prof. Huang, W.C. was the second most productive author, whose studies not only focus on the response of the gut microbiome to endurance exercise [54,55,56], but also investigate the effect of the gut microbiome on exercise physiological adaptation and performance [57,58,59]. Co-authorship analysis is a useful method for identifying existing partnerships and facilitating the development of potential partners [47]. Author analysis showed that 189 authors published more than two papers, but only 22 of them were connected with each other, mainly consisting of authors from the University of Illinois, Mayo Clinic, and the University of Missouri. Most of the collaborations between authors are within institutions.

Most of the research on exercise and the gut microbiome was published in *Nutrients* (IF = 6.706, Q1), indicating that it is currently the most popular journal in this field. Among the journals, *New England Journal of Medicine* had the highest impact factor (IF = 176.08, Q1). Most of the co-cited journals are high-impact Q1 journals, which provide support for research on exercise and the gut microbiome.

### 4.2. Hotspots and Frontiers

Keywords are important indicators in scientific research, reflecting the core content of the relevant study. Keyword analysis could show the closeness and prevalence of the research topics, and help us to quickly capture the distribution and evolution of hotspots in the research field of exercise and the gut microbiome [60]. The cloud map of the top 100 keywords shows that “exercise”, “health”, “diet”, “inflammation”, “intestinal microbiota”, “obesity”, “physical-activity”, “gut microbiota”, “chain fatty-acids”, and “diversity” were the most frequent keywords.

Gut microbial communities are imperative for regulating host digestion, metabolic function, resistance to infection, oxidative stress, hydration status, systematic inflammatory responses, and immune system maturation and functionality [10,61]. Accumulating evidence has suggested that the gut microbiome can be influenced by many environmental factors, such as diet, stress, and exercise [62,63]. In recent years, the bidirectional association between exercise and the gut microbiome has gained increasing attention [64,65].

#### 4.2.1. Effect of Exercise on the Gut Microbiome

##### Cross-Sectional Studies

According to the classification of the study design, studies investigating the effect of exercise on the gut microbiome could be divided into cross-sectional studies and longitudinal studies. Through comparing the differences in the gut microbiome in different populations, cross-sectional studies provide clues for the association between exercise and the microbiome. In our study, the most cited study published in 2014, which is the year in which the studies commonly considered to be the most important and influential in the field were published, analyzed the microbiota composition of athletes and healthy non-athletes [33]. They found that athletes had a higher diversity of gut microorganisms, representing 22 distinct phyla, which in turn positively correlated with protein consumption and creatine kinase [33]. This is the first study to show that exercise increases gut microbial diversity in humans, laying the foundation for research in this field. Recently, some studies have explored the differences in the gut microbiome between athletes and sedentary people or non-athletes, which undoubtedly emphasize the effects of exercise [5,50]. Han, M.Z. et al. [66] compared the differences in the gut microbiome among 19 professional female athletes and 6 young non-athletes. They found that the microbial diversity, taxonomical, functional, and compositional, was significantly different among different people. Both Han, M.Z. et al. [66] and Fontana, F. et al. [67] found that the short-chain fatty acids (SCFAs)-producing bacteria, such as *Clostridiales*, *Eubacterium*, *Blautia, Ruminococcaceae*, and *Faecalibacterium*, were statistically associated with athletes’ samples. Conversely, inflammation-related bacteria, such as *Bilophila* and *Faecalicoccus*, were enriched in physically inactive young adults [68]. Elite athletes have a higher gut microbial diversity, shifted toward bacterial species involved in amino acid biosynthesis and carbohydrate/fiber metabolism, consequently producing key metabolites such as bile acids, SCFAs, and tryptophan [69,70,71]. In addition, athletes have lower levels of circulating bacterial endotoxin lipopolysaccharide and a greater heat shock protein response to heat stress at rest, compared to sedentary individuals [16,72,73]. A higher level of heat shock proteins in the gut may protect the tight junction proteins between epithelial cells, which can improve the long-term stability of the gut barrier [74]. Furthermore, O’Donovan, C.M. et al. [50] revealed the differences in the gut microbiome among athletes in different sports. They found that *Streptococcus suis*, *Clostridium bolteae*, *Lactobacillus phage LfeInf*, and *Anaerostipes hadrus* were associated with a moderate dynamic component, including sports such as fencing, while *Bifidobacterium animalis*, *Lactobacillus acidophilus*, *Prevotella intermedia*, and *F. prausnitzii* were associated with the high dynamic and low static components, including sports such as field hockey [50]. The advances in cross-sectional studies are summarized in Table 6.

##### Longitudinal Studies

Different from cross-sectional studies, longitudinal studies in the field of the gut microbiome and exercise compared the differences in the gut microbiome before and after exercise, intuitively revealing the impact of exercise on the gut microbiome. In fact, the impact of exercise on the gut microbiome is a research hotspot in this field and was the main research topic from 2014 to 2017 according to the Sankey diagram. The keyword co-occurrence analysis showed that cluster 1 and cluster 3 contained keywords associated with the interaction between exercise and the gut microbiome, as well as the physiological and pathological processes of the host. Exercise can have several impacts on the gut microbiome. The composition and diversity of the gut microbiota are changed after exercise, which can influence metabolic profile and immunological responses [14,81]. Skeletal muscle can act as an endocrine organ and release myokines such as irisin and myonectin into the bloodstream, which may act on appetite and lead to changes in the gut microbiome [82,83]. For example, some previous studies have revealed that the abundance of *A. muciniphila* and *Faecalibacterium prausnitzii* increased after exercise [82,84]. Zhong, F. et al. [85] conducted a 60 min exercise program among 14 elderly women. After training, the families *Coriobacteriaceae*, *Asaccharobacter*, *Collinsella*, and *Fusicatenibacter* increased, accompanying the improvement of skeletal muscle mass and fasting blood glucose level [85]. Fernández, J. et al. [86] investigated the effect of four-week endurance/resistance training on the gut microbiome of 8-week-old C57BL6N male mice, demonstrating that anti-inflammatory bacteria, such as *Parabacteroides*, increased after training, while pro-inflammatory bacteria decreased. The latest findings of the effects of exercise training on the gut microbiome are summarized in Table 7. The potential underlying mechanisms are manifold [10,87], including the enhancement of the intestinal barrier integrity [88], modulation of mucosal immunity [89], alteration of intestinal motility and activity of the enteric nervous system [16,90], and repercussions on the intestinal environment in terms of pH, mucus secretion, biofilm formation and availability of nutrients [16,91]. Regular exercise can reduce the heat shock protein’s response to heat stress, protecting tight junction proteins between intestine epithelial cells. Exercise can also modify the gene expression of intraepithelial lymphocytes, decreasing the level of pro-inflammatory cytokines and increasing the level of anti-inflammatory cytokines and antioxidant enzymes [92]. Thus, the exercise represents a hermetic stressor to the gut, that stimulates beneficial adaptations and improves the long-term homeostasis of the gut [16].

The effects of exercise on the gut microbiome depend upon the type of sport, the intensity, and the duration [5,50]. Take endurance sports as an example, both an acute bout of exercise and a long training period can impact the gut microbiome and health [5]. When performing for long durations or in a hot environment, exercise can raise the host’s core temperature and result in heat stress [16,93]. Taking part in 10 min of high-intensity exercise can reduce intestinal blood flow by more than 50% and cause significant gut ischemia [94]. Upon rest, the splanchnic bed undergoes rapid reperfusion. Although the intestine is an anaerobic environment, gut barrier function is transiently impaired during the process of ischemia and post-ischemia reperfusion [94,95]. Exercise-induced heat stress and ischemia may briefly cause more direct contact between the gut mucosal immune system and the microbes residing in the gut lumen and mucosa, with a potential influence on gut microbiome composition [16]. Intense exercise may increase GI epithelial wall permeability and diminish gut mucus thickness, which potentially enables pathogens to enter the bloodstream. Increased pathogens in the bloodstream improve inflammation levels and cause detrimental effects in the host [43]. Conversely, moderate endurance exercise can reduce inflammation and intestinal permeability and improve body composition, leading to positive impacts on gut microbial diversity and human health [53,96]. The impact of exercise type, duration, and intensity on the gut microbiome still needs to be further studied.

According to the thematic evolution analysis, the study of the relationship between disease, the gut microbiome, and exercise occurs throughout the development of this field. As a lifestyle intervention, exercise has been an effective therapy for some metabolic syndromes, cardiovascular diseases, and neuropsychiatric diseases [19,20,21,22]. Recently, the relationship between exercise, the gut microbiome, and disease has gradually become a research hotspot. According to the publications, we summarized the most advanced research progress on this topic in Table 8. In our keyword co-occurrence analysis, both cluster 4 and cluster 5 comprised items that represented the relationship between exercise, the gut microbiome, and disease. Obesity has been reported to be related to many metabolic diseases, such as nonalcoholic fatty liver disease (NAFLD), hypertension, and diabetes [97,98,99]. Some studies have demonstrated the existence of a deleterious microbiota profile in obesity, which can be positively modulated by exercise intervention [100,101]. Quiroga, R. et al. [102] found that exercise can significantly reduce the *Proteobacteria* phylum and *Gammaproteobacteria* class, while increasing some genera, such as *Blautia*, *Dialister*, and *Roseburia*, leading to a microbiota profile similar to that of healthy children. Huang, J. H. et al. [103] showed that the ratio of intestinal microbiota, *Firmicutes* to *Bacteroidetes*, which could potentially contribute to promoting weight loss, displayed a marked increase after exercise and diet intervention. For patients with insulin resistance, exercise can reduce intestinal inflammation and modify the profile of the gut microbiome, such as the genera *Clostridium* and *Blautia*, which are closely correlated with improvements in glucose homeostasis and insulin sensitivity [104]. These results highlight the value of exercise intervention as an efficient nonpharmacological therapy in metabolic endocrine diseases. In addition, the role of exercise as a kind of treatment for cancer patients, is gradually gaining increasing attention [105]. As a non-pharmaceutical intervention, exercise may be beneficial in relieving adverse reactions during cancer treatment, promoting rehabilitation, and improving prognosis [106,107,108,109]. The mechanisms are still unknown. Some researchers have found connections between the gut microbiome and exercise as a non-pharmaceutical treatment. Sampsell, K. et al. demonstrated that the post-exercise gut microbiota from an individual who demonstrated a positive microbial response to exercise, significantly altered the tumor microenvironment and gut microbial response to chemotherapy [110]. Wang, W. Y. et al. showed that swimming pretreatment can protect mice from colitis-associated cancer, by intervention in the possible link between colonic lipid metabolites and PGE2/EP2 signaling [111]. However, there is still a lack of research in this field. Deep investigation of the relationship between exercise, the gut microbiome, and cancer would be beneficial to further optimize the methods of exercise intervention and expand their application in cancer treatment.

**Table 7 microorganisms-11-00903-t007:** Longitudinal studies investigating the effects of exercise on the gut microbiome.

Author	Publication Year	Subjects	Exercise	Effects on Composition and/or Metabolite of Gut Microbiota	Physiological Metabolite/ Functional Change
Wang, R.H. et al. [112]	2023	30 young adolescents aged 12–14 years	Moderate-intensity exercise, comprised of 30 min of running per day, 4 days a week for 3 months	**After training:** **Genus:** *Coprococcus* ↑, *Blautia* ↑, *Dorea* ↑, *Tyzzerella* ↑ **Species:** *Tyzzerella nexilis* ↑, *Ruminococcus obeum* ↑	The improvement of depressive symptoms ↑, KEGG pathways belonging to the neurodegenerative diseases ↓
Wang, Y.J. et al. [113]	2023	Chronic mouse model of Parkinson’s disease, induced by MPTP	Rotarod walking training (5 times a week at 25 rpm for 20 min for 8 weeks)	**After training:** **Family:** *Oscillospiraceae* ↑, *Ruminococcaceae*↑ **Genus:** *Lachnospiraceae_NK4A136_*group ↑, *unclassfied_f__Oscillospiraceae* ↑, *Alloprevotella* ↓	DA and TH in the striatum ↑, BDNF gene expression ↑, IL-1β gene expression in the striatum ↓
Craven, J. et al. [114]	2021	Middle-distance runners	Three weeks of NormTr + three weeks of HVolTr + a one-week TaperTr	**Following HVolTr:** *Pasterellaceae* ↓, *Lachnoclostridium* ↓, *Haemophilus* ↓, *S. parasagunis* ↓, and *H. parainfluenzae* ↓, *R. callidus* ↑	-
Bielik, V. et al. [115]	2022	Young competitive male and female swimmers	7-week high-intensity swimming training program	**HIT group:** *Firmicutes* ↓, *Bacteroidota* ↑, *Actinobacteriota* ↑ **HITB group:** *Firmicutes* ↓, *Bacteroidota* ↑, *Actinobacteriota* ↓	Uric acid ↓, alanine transaminase ↓, vitamin D ↑, lactate ↑, pyruvate ↑, acetate ↓, butyrate ↓
Zhong, F. et al. [85]	2022	14 elderly women	60 min exercise program: 10 min warm-up + 20 min aerobic exercise (20 min) + 25 min resistance exercise + 5 min cool down	**After training:** **Order:** *Coriobacteriales* ↑, **Family:** *Coriobacteriaceae* ↑, *Asaccharobacter* ↑, *Collinsella* ↑, *Fusicatenibacter* ↑	The skeletal muscle mass ↑, fasting blood glucose ↓, CVD risk ↓
Qiu, L.W. et al. [116]	2022	22 university students with/without sleep disorders	Jog three times per week, at a speed of 8–9 km/h, for a distance of 4–7 km each time	**After training:** *Blautia* ↑, *Eubacterium hallii*↑, *Agathobacter* ↓	Sleep quality ↑
Dupuit, M. et al. [117]	2022	Postmenopausal women who were overweight or obese	HIIT + RT (3 times per week for 12 weeks)	**FM ↑:** *Bifidobacteriaceae* ↑, *Paraprevotellaceae* ↓, *Prevotellaceae* ↓ **Muscle mass ↑:** *Bifidobacteriaceae* ↓, *Paraprevotellaceae* ↑, *Prevotellaceae* ↑ **HDL-C ↑:** *Bifidobacteriaceae* ↓, *Paraprevotellaceae* ↓, *Prevotellaceae* ↓, *Streptococcaceae* ↓, *Desulfovibrionaceae* ↓	Physical fitness (maximal oxygen consumption, peak power output) ↑, segmental muscle mass ↑, total abdominal and visceral fat mass ↓
Sato, M. et al. [118]	2022	Ultramarathon runners	Run 96.102 km or 99.12 km within 38–44 h	**After training:** Butyrate-producing bacteria, such as *F. prausnitzii*↓, *C. aerofaciens* ↑	Butyrate levels in the intestine ↓
Fernández, J. et al. [86]	2021	8-week-old C57BL6N male mice	Four-week endurance training/resistance training	**After training:** *Ruminococcus gnavus* ↓, *Parabacteroides* ↑ **Endurance performance ↑:** *Lachnospiraceae* family ↓, *Lactobacillaceae* family ↓, *Prevotellaceae* family ↑, *Prevotella* genus ↑, *Akkermansia muciniphila* ↑ **Resistance performance ↑:** *Desulfovibrio sp.* ↓, *Proteobacteria taxon* ↓, *Alistipes* ↑	H_2_S production by *Desulfovibrio* ↑, anti-inflammatory *Parabacteroides* ↑, pro-inflammatory bacterium ↓
Chen, H. et al. [119]	2021	3–4-week-old C57BL/6J WT female mice	Four-week SE training + 7 days post-immunization SE training	**After training:** *Anaerotruncus* ↓, *Jeotgalicoccus* ↓, *Anaerotruncus* ↓, *Alistipes* ↓, *Ruminococcus* ↓, *Desulfovibrio* ↓, *Clostridium* ↑, *Parabacteroides* ↑, *Christensenella* ↑, *Dorea* ↑, *Roseburia* ↑, *Paraprevotella* ↑	Th17 responses ↓, Treg responses ↑, intestinal mucosal permeability ↓, MS disease severity ↓, neuropathology scores ↓
Moitinho-Silva, L. et al. [120]	2021	42 healthy physically inactive German male and female volunteers aged 20 to 45 years	Endurance training, strength training	**After training:** *Coprococcus* genera ↑, *Parasutterella* genera ↑, *Ruminococcaceae* family ↑, *Dialister* genera ↓, *Odoribacter* ↓, *Phascolarctobacterium* ↓	**Strength intervention group:** Lymphocytes ↑, MCHC ↓ **Endurance group:** Hip circumference ↓, physical working capacity ↑
Resende, A.S. et al. [121]	2021	24 previously sedentary men	10-week moderate aerobic exercise, 150 min per week of supervised moderate (60–65% of VO_2_ peak) aerobic exercise	**After training:** *Streptococcus* genus ↑, *Clostridiales*-order genus ↓ **VO_2_ peak ↑:** *Odoribacter* ↑, *Roseburia* ↑, *Sutterella* ↑ **BMI ↑:** *Desulfovibrio* ↓ **Body fat ↑:** *Faecalibacterium* ↓	VO_2_ peak ↑, Wpeak ↑, AT1 ↑, cardiorespiratory fitness ↑
Li, K.F. et al. [122]	2021	54 male C57BL/6J mice at 12 weeks of age	4-week wheel-running exercise	**After training:** Gut microbial diversity ↑ **Phyla:** *Firmicutes* ↓, *Proteobacteria* ↓, *Bacteriodetes* ↑, *Firmicutes*/ *Bacteriodetes* ratio ↓ **Family:** *Bacteroidales_S24-7* ↑, *Prevotellaceae* ↑, *Desulfovibrionaceae* ↓, *Peptostreptococcaceae* ↓	LPS levels in the blood ↓, synovial fluid ↓, TLR4 and MMP-13 expression levels ↓, cartilage degeneration ↓
Verheggen, R.J.H.M. et al. [123]	2021	20 inactive participants with obesity (BMI > 30 kg/m^2^)	8-week exercise intervention (2 to 4 times per week, on 65–85% of heart rate reserve).	**After training:** **Genus:** *Ruminococcus gauvreauii* ↑, *Lachnospiraceae* FCS020 ↑, *Anaerostipes* ↑ **M-value ↑:** *R. gauvreauii* ↑ **VO_2_ max ↑:** *R. gauvreauii* ↑, *Anaerostipes* ↑	Insulin sensitivity ↑, visceral adiposity ↓, cardiorespiratory fitness levels ↑
Ahrens, A.P. et al. [124]	2021	73 adults (37 male, 36 female)	One-week, in-house, lifestyle-based “immersion program”, including dedicated fitness	**After training:** *Lachnospiraceae* ↑, *Oscillospirales* ↑, *Ruminococcaceae* ↑, *Faecalibacterium* ↑, *Roseburia* ↑, *Blautia* ↑, *Anaerostipes* ↑, *Subdoligranulum* ↑, *Bacteroidaceae* ↓, *Phascolarctobacterium* ↓	Blood pressure ↓, total cholesterol ↓, triglycerides ↓, cardiovascular risk ↓

**Abbreviations:** AT1: anaerobic threshold 1; BNDF: brain-derived neurotrophic factor; CVD: cardiovascular disease; DA: dopamine; HIT: high-intensity training; HITB: high-intensity training and use of probiotic cheese; HIIT + RT: high-intensity interval training and resistance training; HVolTr: high-volume running training; LPS: lipopolysaccharide; KEGG: Kyoto Encyclopedia of Genes and Genomes; MCHC: mean corpuscular hemoglobin concentration; MMP: metalloproteinase; MPTP: 1-methyl-4-phenyl-1,2,3,6-tetrahydropyridine; MS: multiple sclerosis; NormTr: normal running training; SE: strength exercise; TaperTr: taper training; TH: tyrosine hydroxylase; TLR4: Toll-like receptor 4; ↑: increase; ↓: decrease.

**Table 8 microorganisms-11-00903-t008:** Advances in the relationship between exercise, the gut microbiome, and disease.

Related Diseases		Author	Publication Year	Objects (Materials)	Results
Cancer	Breast cancer	Sampsell, K. et al. [110]	2022	Breast cancer patient survivors, C57BL/6 mice	Exercise and prebiotic fiber demonstrated adjuvant action, potentially via an enhanced anti-tumor immune response modulated by advantageous gut microbial shifts.
	Colitis-associated cancer (CAC)	Wang, W.Y. et al. [111]	2022	C57BL/6 mice	Swimming pretreatment can protect mice from CAC, possibly through regulating gut microbiota and intestinal SCFAs, and affecting the function of colonic lipid metabolites and choline metabolism in cancer.
	Lung cancer	Marfil-Sánchez, A. et al. [125]	2021	Early-stage lung cancer patients	There are some beneficial bacterial species for improving recovery of overall physical condition and lung capacity in patients one year after lung resection surgery. Some bacterial metabolic pathways might be associated with increased oxygen uptake and exercise tolerance.
	Prostate cancer	Frugé, A.D. et al. [126]	2020	Prostate cancer patients	Relative abundance of *Bifidobacterium* was associated with *Cathepsin L (CTSL)* and free fatty acids (FFAs); Firmicutes was positively related with change in physical activity. Although glucose metabolism improved, it was inversely related with Ki67 and *CTSL*.
Metabolic endocrine diseases	Obesity	Verheggen, R. et al. [123]	2021	Obese patients	Eight-week exercise training in obese humans leads to marked improvements in insulin sensitivity and body composition, accompanied by modest changes in *Ruminococcus gauvreauii*, *Lachnospiraceae FCS020* group, and *Anaerostipes*, which all belong to the *Firmicutes* phylum.
	Childhood obesity	Quiroga, R. et al. [102]	2020	Obese pediatric patients	Exercise training could be considered an efficient nonpharmacological therapy, reducing inflammatory signaling pathways induced by obesity in children via microbiota modulation. The existence of obesity-related deleterious microbiota profiles can be positively modified by exercise.
	Nonalcoholic fatty liver disease (NAFLD)	Cheng, R.T. et al. [99]	2022	NAFLD and prediabetes patients	Combined aerobic exercise and diet intervention are associated with diversified and stabilized keystone taxa. A personalized gut microbial network at baseline could predict the individual responses in liver fat, to exercise intervention.
	Obesity, insulin resistance, obstructive sleep apnea	Khalyfa, A. et al. [127]	2021	C57BL/6 mice	Intermittent hypoxia (IH) exposures induce changes in gut microbiota (GM), increase gut permeability, and alter plasma exosome cargo, the latter inducing adipocyte dysfunction (increased insulin resistance). GM alterations can be improved with physical activity.
	Prediabetes	Liu, Y. et al. [104]	2020	Medication-naive men with prediabetes, C57BL/6J mice	Exercise-induced alterations in the gut microbiota correlated closely with improvements in glucose homeostasis and insulin sensitivity. The microbiome of responders exhibited an enhanced capacity for biosynthesis of short-chain fatty acids and catabolism of branched-chain amino acids.
Immune diseases	Multiple sclerosis	Mokhtarzade, M. et al. [128]	2021	Multiple sclerosis (MS) patients	Home-based exercise significantly increased *prevotella* counts, and decreased *akkermansia muciniphila* counts, which can probably have a beneficial effect on MS disease pathology and course. These changes were associated with changes in IL-10.
	Celiac disease	Warbeck, C. et al. [129]	2021	Adults with celiac disease	Following 12-week high-intensity interval training (HIIT) plus lifestyle education intervention, the group showed beneficial changes in the gut microbiota of adults with celiac disease.
Neurological and psychotic disorders	Insomnia symptoms	Magzal, F. et al. [130]	2022	Older adults with insomnia	Different microbiota taxa in each physical activity group, increased SCFAs in feces of less active individuals, and significant associations among physical activity, gut microbiota, SCFAs, and sleep parameters were observed.
	Alzheimer’s disease	Abraham, D. et al. [131]	2019	APP/PS1 transgenic (APP/PS1^TG^) mice	Exercise and probiotic treatment can decrease the progress of Alzheimer’s disease and the beneficial effects could be partly mediated by alteration of the microbiome.

Abbreviations: CAC: colitis-associated cancer, CTSL: Cathepsin L, FFAs: free fatty acids, GM: gut microbiota, HIIT: high-intensity interval training, IH: intermittent hypoxia, NAFLD: nonalcoholic fatty liver disease, SCFAs: short-chain fatty acids.

#### 4.2.2. Relationship between Exercise and Gut-Brain Axis (GBA)

The enteric (ENS) and central nervous systems (CNS) have identical tissue derivation. The bidirectional communication between ENS and CNS, which is known as the gut–brain axis (GBA), has embryogenic origins [132]. Neuroendocrine, immune, and neuronal regulation mediate communication [1,133]. Bidirectional signal traveling is mainly mediated by several molecules produced by the gut microbiome, such as SCFAs and tryptophan, which interact with enteroendocrine cells and activate the vagus nerve [1,134]. The alterations in the gut microbiome could exert an influence on CNS through local stimulation of neurons, direct release of microbiome-produced molecules to the bloodstream, and the modulation of the immune system [135,136,137]. Recent evidence suggested that exercise could also influence the GBA through regulating the gut microbiome [138]. Several metabolites produced by the gut microbiome, such as SCFAs and bile acids, may interact with enteroendocrine cells, by propagating a long-distance signal [134]. For example, exercise could increase the relative abundance of the family *Lachnospiraceae*, which is capable of producing butyrate. Butyrate can over-regulate brain-derived neurotrophic factor (BDNF) expression in the hippocampus and frontal cortex, support the survival of existing neurons, and stimulate the formation of new neurons and synapses [139]. Dohnalová, L. et al. [140] reported a gut–brain connection in mice, that enhances exercise performance by augmenting the dopamine signal. The gut microbiome could generate intestinal fatty acid amide metabolites, that stimulate TRPV1-expressing sensory neurons, which in turn send an exercise-induced afferent signal to the brain and promote the downregulation of expression of the enzyme monoamine oxidase (MAO) in the striatum. Thereby the decrease in MAO elevates dopamine levels in the ventral striatum during exercise and enhances exercise capability [140]. When the gut microbiome is depleted, spinal afferent neurons are ablated, or dopamine is blocked, exercise capacity is inhibited [140]. Their findings indicated that the rewarding properties of exercise are influenced by gut-derived interoceptive circuits, and provided a microbiome-dependent explanation for inter-individual variability in exercise performance [140].

Furthermore, disruption of the gut barrier and gut dysbiosis have been considered one of the mechanisms of major depression and several neurodegenerative diseases, with the GBA playing a critical role [1,141]. It has been observed that microbial diversity was reduced in Alzheimer’s disease patient feces, compared to the general population. A decrease in the abundance of *Firmicutes* and *Actinobacteria* and an increase in the abundance of *Bacteroidetes* was observed at the phylum level [142,143]. Amyloid deposition was related to an increased stool content of the pro-inflammatory taxa *Escherichia*/*Shigella* and low content of the anti-inflammatory taxon *Eubacterium rectale* [144]. Abraham et al. found that after 20 weeks of treadmill exercise, the spatial memory of transgenic mice significantly improved, and the level of β-amyloid plaques was reduced, showing the potential ability of exercise to improve the cognitive function of Alzheimer’s disease patients [142]. An increase in microglia near β-amyloid plaques was observed, which plays an important role in brain development, neuron support, and repair, underlining the neuroprotective effect of exercise [1,145]. These effects may be associated with the abundance of some bacterial strains involved in disease exacerbation. Exercise may regulate cognitive conditions and functionality through regulation of the microbiome composition and subsequently the generation of protective molecules [1]. However, only a few studies have investigated the neural aspect of the association between exercise and the gut microbiome. Future studies should explore this field more deeply.

#### 4.2.3. Impact of Gut Microbiome on Exercise

The gut microbiome can act as an endocrine organ and regulates numerous facets of human biology [39]. The gut microbiome and its metabolites likely indirectly influence athlete health, training, sports performance, and post-exercise recovery [146]. The modulation of the gut microbiota and its metabolites, such as SCFAs, including butyrate and propionate, bile acids, and tryptophan, which are beneficial for the organism, can allow athletes to conduct huge volumes of training or to improve their sports performance [5]. For example, SCFAs may reduce the permeability of the intestinal barrier and decrease the production of inflammatory cytokines, thus helping to alleviate gastrointestinal discomfort, reduce symptoms of fatigue, enhance skeletal muscle mass and function, and improve athletic performance [147,148,149]. The advances in the understanding of the impact of the gut microbiome on exercise are shown in Table 9. Some evidence can support the impact of the gut microbiome on exercise. Kim, K.H. et al. [150] transferred the gut microbiome from young mice to old mice. They observed that after the transfer, the old mice’s muscle fiber thickness, grip strength increased, and the water retention ability of the skin was enhanced, with a thickened stratum corneum. This result indicated that the gut microbiome from the young mice rejuvenated the physical fitness of the old mice, by altering the microbial composition of the gut [150]. Moreover, in the keyword co-occurrence analysis, the timing diagram showed that the impact of prebiotic and/or probiotic supplementation on sports performance among athletes gradually became a research hotspot. The majority of probiotic strains belong to the *Bifidobacterium*, *Lactobacillus*, *Enterococcus*, or *Propionibacterium* genera or certain yeasts [151,152]. Through their potential role in impacting tight junction proteins, modulating the immune system, and inhibiting pathogen colonization, probiotics can help to ease or prevent certain GI symptoms or disorders associated with extreme exercise [153,154,155]. Probiotic supplement-induced changes in the gut microbiome may help with the brief immunosuppression period, influence the status of host energy metabolism, and reduce susceptibility to infections, which is beneficial to athletes’ health [115,153,156]. Santibañez-Gutierrez, A. et al. [157] demonstrated that probiotic supplementation exerts a positive effect on performance, with aerobic metabolism predominance, in a trained population, particularly when the supplementation period was ≤4 weeks and single strain probiotics were consumed. Salleh, R.M. et al. [158] provided probiotic supplements containing *Lactobacillus casei* Shirota to 30 university badminton players. They found that probiotic supplementation improved aerobic capacity and relieved anxiety and stress, but did not influence the speed, strength, leg power, and agility [158]. However, there is still not sufficient evidence to support the hypothesis that probiotics can directly improve exercise performance. Further investigations are warranted.

### 4.3. Limitations

There are also some limitations in this study. First, we only searched for publications from a single source (WOSCC database). Although using the WOSCC database to conduct bibliometric analysis has been widely accepted by researchers, the omission of publications from other sources may influence the results of this study. In addition, there are some biases that may affect the analysis results, such as selection bias originating from the process of manually screening papers, and publication bias.

## 5. Conclusions

In conclusion, by means of bibliometric analysis, our study reveals current research trends and hotspots at a global level, and provides an objective assessment of the contributions of academic groups and individual researchers. The number of publications on exercise and the gut microbiome has shown an increasing trend over the last 20 years. The leading countries/regions in this field are the USA, China, and Europe. Most of the active institutions with high production and excellent academic impact were from Europe and America. *Nutrients* was the most productive journal in this field. Prof. Cotter, P.D. was the most influential author. However, cooperation and communication among various countries/regions, institutions, and authors should be strengthened. The relationship between the gut microbiome, exercise, health, and disease, and how the gut microbiome influences exercise performance, are currently two important facets of research. Despite the relevant research achievements in this field during the last two decades, important gaps and unmet needs still exist. Further studies are needed to fully discriminate the mechanisms by which exercise influences the composition and functionality of the gut microbiome, provide further insights into optimal therapies to influence the gut microbiota and its relationship with health and disease, and explore the possibility of improving the efficacy of disease treatment by optimizing exercise intervention and regulating the gut microbiome.

## Figures and Tables

**Figure 1 microorganisms-11-00903-f001:**
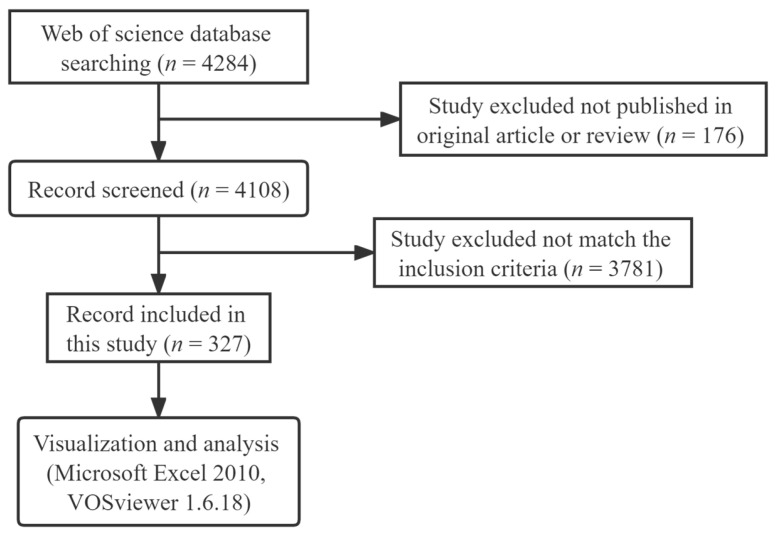
Flowchart of literature selection.

**Figure 2 microorganisms-11-00903-f002:**
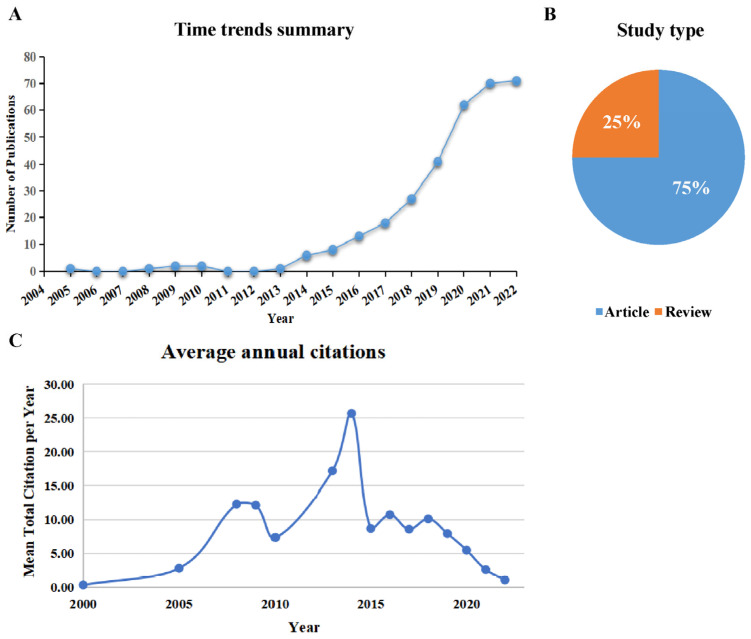
(**A**) Time trends summary. (**B**) Study type composition summary. (**C**) Summary of average annual citations.

**Figure 3 microorganisms-11-00903-f003:**
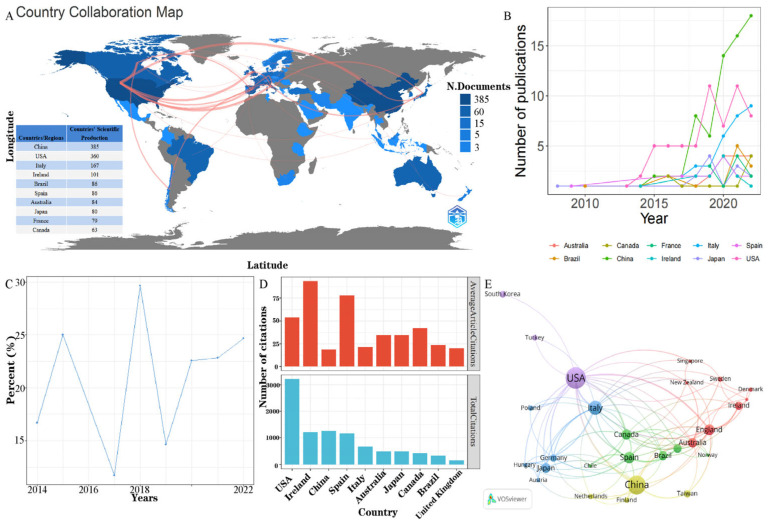
(**A**) Global distribution and international interactions of research in the field of the microbiome and exercise. (**B**) The annual distribution of research in the field of the microbiome and exercise, from the top 10 countries between 2008 and 2022. (**C**) The annual proportion of research in the field of the microbiome and exercise from China. (**D**) Total and average number of article citations in the top ten most highly cited countries. (**E**) Network visualization map of international research collaboration in the microbiome and exercise among the leading active countries.

**Figure 4 microorganisms-11-00903-f004:**
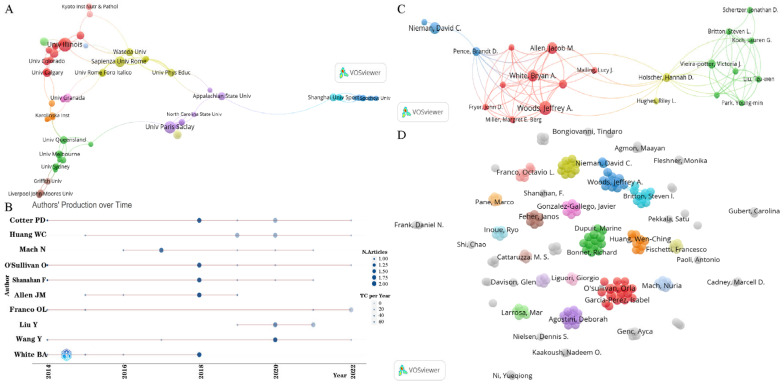
(**A**) Network visualization map of international research collaboration in the microbiome and exercise among the leading active institutions. (**B**) Authors’ production over time, in the field of the microbiome and exercise between 2014 and 2022. (**C**) Network visualization map of authors who connected with each other. (**D**) Network visualization map of all authors.

**Figure 5 microorganisms-11-00903-f005:**
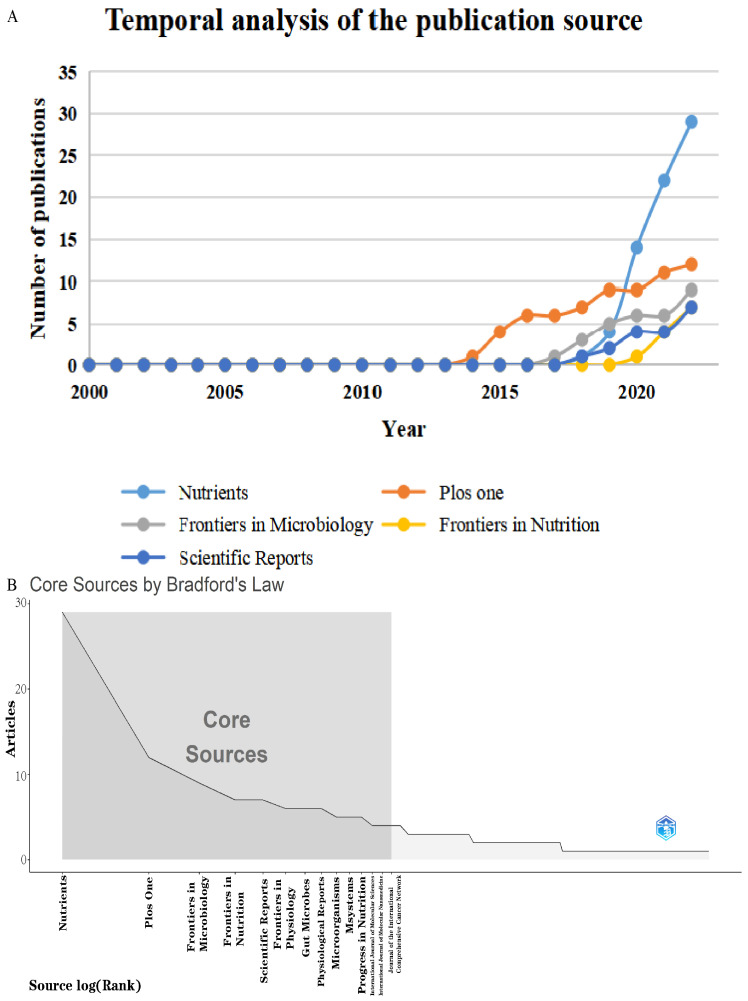
(**A**) Temporal analysis of the publication source. (**B**) Core sources classified by Bradford’s Law.

**Figure 6 microorganisms-11-00903-f006:**
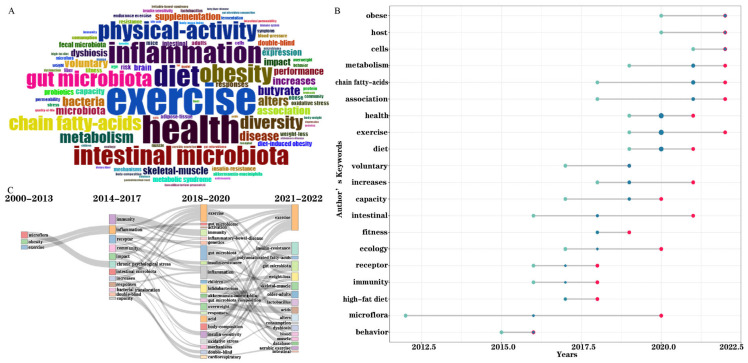
(**A**) Cloud graph of high–frequency keywords in the field of microbiome and exercise. (**B**) Trend topic analysis of keywords. (**C**) Evolution of research topics in the field of the microbiome and exercise.

**Figure 7 microorganisms-11-00903-f007:**
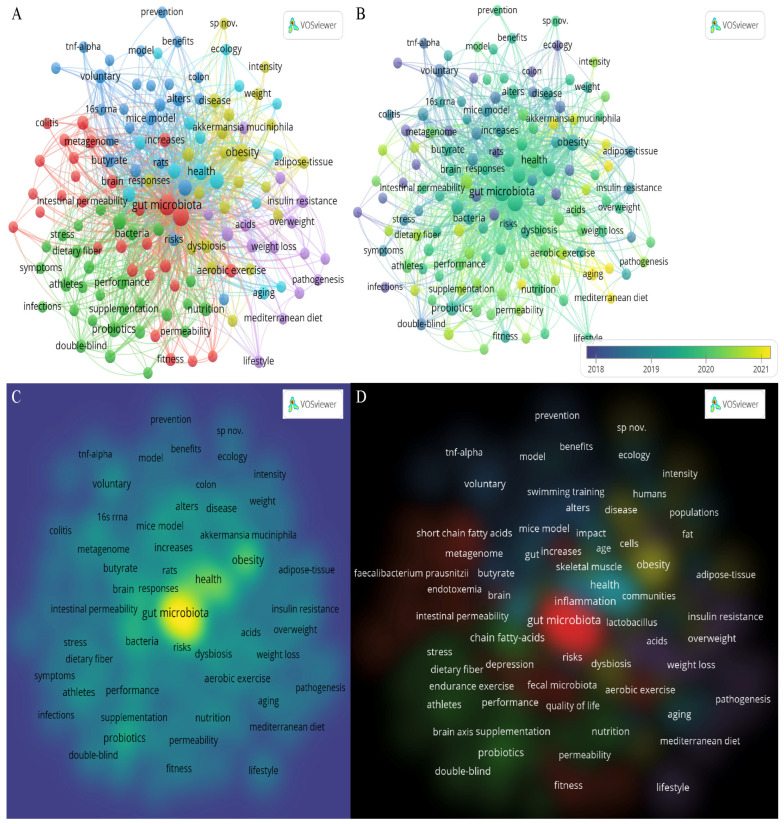
Network visualization map of terms in title/abstract fields of publications related to microbiome and exercise. (**A**) Network visualization. (**B**) Overlay visualization. (**C**) Item density visualization. (**D**) Cluster density visualization.

**Table 1 microorganisms-11-00903-t001:** Top ten countries/regions with the most publications.

No.	Countries/Regions	Publications	Citations	Average Article Citations	h-Index	Total Link Strength
1	USA	81	3218	53.63	30	50
2	China	64	1254	18.72	18	20
3	Italy	38	664	21.42	14	34
4	Spain	26	1166	77.73	14	30
5	England	24	159	19.88	11	32
6	Canada	21	419	41.90	9	30
7	Japan	19	481	34.36	11	13
8	Australia	18	482	34.43	10	15
9	Brazil	17	332	23.71	8	8
10	France	16	130	13.00	8	16

**Table 2 microorganisms-11-00903-t002:** Top ten active institutions with the most publications about the gut microbiome and exercise.

No.	Institutions	Publications	Citations	Time Cited (per Article)	Total Link Strength	Countries/Regions
1	Univ. Illinois	11	1043	95	11	USA
2	Univ. Paris Saclay	9	556	62	10	France
3	Sapienza Univ. Rome	7	138	20	17	Italy
4	Univ. Coll. Cork	7	127	18	9	Canada
5	Teagasc Food Res. Ctr.	6	926	154	11	Ireland
6	Imperial Coll. London	5	362	72	13	England
7	Natl. Taipei Univ. Nursing and Hlth. Sci.	5	83	17	11	Taiwan
8	Natl. Taiwan Sport Univ.	5	173	35	11	Taiwan
9	Univ. Calgary	5	17	3	7	Canada
10	Univ. Catolica Brasilia	5	323	65	12	Brazil

**Table 3 microorganisms-11-00903-t003:** Top ten active authors with the most publications about the gut microbiome and exercise.

No.	Authors	Publications	Total Citations	h-Index	g-Index	m-Index	Total Link Strength
1	Cotter, P.D.	7	1101	5	7	0.500	41
2	Huang, W.C.	6	189	5	6	0.556	11
3	Mach, N.	6	566	6	6	0.750	12
4	O’Sullivan, O.	6	1065	5	6	0.500	43
5	Shanahan, F.	6	1073	5	6	0.500	35
6	Allen, J.M.	5	780	5	5	0.556	19
7	Franco, O.L.	5	339	3	5	0.300	9
8	Liu, Y.	5	191	5	5	1.000	0
9	Wang, Y.	5	654	4	5	0.400	0
10	White, B.A.	5	790	5	5	0.500	22

**Table 4 microorganisms-11-00903-t004:** Top ten journals with the most publications in the field of the gut microbiome and exercise.

Journal	Publications	Impact Factor	Quartile in Category	h-Index	g-Index	m-Index	Total Citations
*Nutrients*	29	6.706	Q1	13	21	2.167	467
*PLoS ONE*	12	3.752	Q2	10	12	1	829
*Frontiers in Microbiology*	9	6.064	Q1	6	9	0.857	421
*Frontiers in Nutrition*	7	6.590	Q1	3	7	0.750	99
*Scientific Report*	7	4.997	Q2	4	7	0.667	125
*Frontiers in Physiology*	6	4.755	Q1	5	6	0.625	356
*Gut Microbes*	6	9.434	Q1	6	6	1	207
*Physiological Reports*	6	0.720	Q3	4	6	0.667	108
*Microorganisms*	5	4.926	Q2	3	5	0.750	45
*Msystems*	5	7.328	Q1	4	5	0.571	155

**Table 5 microorganisms-11-00903-t005:** The top 10 most cited publications in the field of gut microbiome and exercise.

No.	Author	Publication Year	Document Type	Journal	Title	Times Cited in WOSCC Database
1	Siobhan, C.F. et al. [33]	2014	Article	*Gut*	Exercise and associated dietary extremes impact on gut microbial diversity	644
2	Christian, E.C. et al. [44]	2014	Article	*PLoS ONE*	Exercise Prevents Weight Gain and Alters the Gut Microbiota in a Mouse Model of High Fat Diet-Induced Obesity	347
3	Santacruz, A. et al. [43]	2009	Article	*Obesity*	Interplay Between Weight Loss and Gut Microbiota Composition in Overweight Adolescents	317
4	Jacob, A.M. et al. [17]	2018	Article	*Medicine and Science in Sports and Exercise*	Exercise Alters Gut Microbiota Composition and Function in Lean and Obese Humans	279
5	Scheiman, J. et al. [42]	2019	Article	*Nature Medicine*	Meta-omics analysis of elite athletes identifies a performance-enhancing microbe that functions via lactate metabolism	251
6	Barton, W. et al. [41]	2018	Article	*Gut*	The microbiome of professional athletes differs from that of more sedentary subjects in composition and particularly at the functional metabolic level	236
7	Monda, V. et al. [15]	2017	Review	*Oxidative Medicine and Cellular Longevity*	Exercise Modifies the Gut Microbiota with Positive Health Effects	213
8	Estaki, M. et al. [40]	2016	Article	*Microbiome*	Cardiorespiratory fitness as a predictor of intestinal microbial diversity and distinct metagenomic functions	199
9	Clark, A. et al. [39]	2016	Review	*Journal of the International Society of Sports Nutrition*	Exercise-induced stress behavior, gut-microbiota-brain axis and diet: a systematic review for athletes	187
10	Jeong, C.J. et al. [38]	2013	Article	*Environmental Health Perspectives*	Exercise Attenuates PCB-Induced Changes in the Mouse Gut Microbiome	187

**Table 6 microorganisms-11-00903-t006:** Cross-sectional studies in the field of the gut microbiome and exercise.

Author	Publication Year	Subjects	Results
Fontana, F. et al. [67]	2023	254 healthy athletes vs. 164 healthy sedentary adults	SCFAs microbial producers including *Faecalibacterium*, *Eubacterium*, *Blautia*, and *Ruminococcus* species, are statistically associated with athletes’ samples. The EFC related to athletes was positively linked to 752 enzymes (EC numbers) and 73 HIBS. The EFC related to sedentary adults was positively linked only to 105 EC numbers and 14 HIBS.
Kulecka, M. et al. [75]	2023	109 well-characterized Polish male e-sports players, 25 endurance athletes, 36 healthy students of physical education	The differences of lifestyle and dietary habits appear to have little effect on most gut microbiota parameters. Several metabolic pathways including fermentation, amino acid biosynthesis and degradation, carbohydrate biosynthesis and degradation, fatty acid biosynthesis and degradation, and the TCA cycle pathways, were over-represented in professional athletes compared with e-sports players and students.
Hintikka, J.E. et al. [76]	2022	27 national team cross-country skiers vs. 27 normally physically active people	Phylogenetic diversity and the abundance of mucin-degrading gut microbial taxa were lower among athletes. *Butyricicoccus* is associated positively with HDL cholesterol, HDL2 cholesterol, and HDL particle size. The *R. torques* group was less abundant in the athlete group and positively associated with total cholesterol and VLDL and LDL particles.
Xu, Y.J. et al. [68]	2022	22 elite athletes vs. 44 general young adults	No significant difference was observed in both alpha and beta diversity. Compared to general young adults, elite athletes had a significantly higher abundance of *Clostridiaceae* and *Megamonas_rupellensis*. Inflammation-related bacteria, such as *Bilophila* and *Faecalicoccus,* were enriched in physically inactive young adults compared to two other groups.
Šoltys, K. et al. [77]	2021	13 elderly endurance athletes vs. 9 healthy controls	Lifelong endurance training does not bring significant benefit regarding overall gut microbiota. Continual exercise by elderly endurance athletes is associated with favorable gut microbiota composition on the lower taxonomy level. The *Bacteroides* to *Prevotella* ratio seems to distinguish the endurance trained elderly from healthy controls.
Babszky, G. et al. [78]	2021	20 actively competing athletes vs. 20 sedentary subjects	No significant differences in the fecal microbiome flora of trained and sedentary subjects who were diagnosed as positive for COVID-19 was detected. The levels of *Bacteroidetes* were enhanced during mild COVID-19 infection, supporting the immune system by suppressing the activation of TLR4 and ACE2 receptors.
Kulecka, M. et al. [79]	2020	25 endurance athletes (14 marathon runners and 11 cross-country skiers) vs. 46 sedentary healthy controls	Excessive training is associated with both large alterations to microbial community composition and promotion of higher bacterial diversity. Endurance athletes presented a lowered abundance of major gut microbiota genus, *Bacteroidetes* and higher abundance of complex carbohydrates fermenters (from *Prevotella* genus and *Firmicutes* phylum).
Han, M.Z. et al. [66]	2020	19 professional female rowing athletes (7 AE + 6 YE) vs. 6 YN	The microbial diversities, taxonomical, functional, and phenotypic compositions, of AE, YE, and YN were significantly different. SCFA-producing bacteria, such as *Clostridiales*, *Ruminococcaceae* and *Faecalibacterium*, are dominant in the microbial community of elite athletes. ATP metabolism, multiple sugar transport systems, and carbohydrate metabolism are enriched in the microbial community of elite athletes.
O’Donovan, C.M. et al. [50]	2020	37 elite Irish athletes across 16 different sports	*Streptococcus suis*, *Clostridium bolteae*, *Lactobacillus phage LfeInf*, and *Anaerostipes hadrus* were found to be associated with a moderate dynamic component, including sports such as fencing. *Bifidobacterium animalis*, *Lactobacillus acidophilus*, *Prevotella intermedia* and *F. prausnitzii* were found to be associated with the high dynamic and low static components, including sports such as field hockey, while *Bacteroides caccae* was found to be associated with the high dynamic and static components, including sports such as rowing.
Fart, F. et al. [80]	2020	28 physically active senior orienteering athletes vs. 70 community-dwelling older adults and	Compared to the community-dwelling older adults, the physically active senior orienteers had a more homogeneous microbiota within the group, a higher abundance of *Faecalibacterium prausnitzii*, and a lower abundance of *Parasutterella excrementihominis* and *Bilophila* unclassified.

Abbreviations: ACE: angiotensin converting enzyme; AE: adult elite athletes; EFC: enzymatic functional clusters; HDL: high-density lipoprotein; HIBS: high-biological-impact synthases; LDL: low-density lipoprotein; SCFA: short-chain fatty acid; SCG: sports classification groups; TCA: tricarboxylic acid cycle; TLR4: Toll-like receptor 4; VLDL: very low-density lipoprotein; YE: youth elite athletes; YN: youth non-elite athletes.

**Table 9 microorganisms-11-00903-t009:** Impact of the gut microbiome on exercise.

Author	Publication Year	Subjects	Intervention	Results
Kim, K.H. et al. [150]	2022	5-week-old young mice, 12-month-old mice	Transfer the gut microbiota from the young mice to the old mice	The young-mice-derived gut microbiota rejuvenated the physical fitness of the old mice, by altering the microbial composition of the gut and gene expression in muscle and skin. After the transfer, the old mice’s muscle fiber thickness, grip strength increased, and the water retention ability of the skin was enhanced, with a thickened stratum corneum. The host fitness of the old mice significantly improved.
Furber, M.J.W. et al. [159]	2022	16 highly trained endurance runners	Isocaloric HPD or HCD	HCD improved time-trial performance by 16.5%, and was associated with expansion of *Ruminococcus* and *Collinsella bacterial* spp. HPD led to a reduction in performance by 223.3%, accompanied by significantly reduced diversity and altered composition of the gut phageome, as well as enrichment of both free and inducible *Sk1virus* and *Leuconostoc* bacterial populations. Gut microbial stability during acute dietary periodization was associated with greater athletic performance.
Liu, G. et al. [160]	2022	7 health professional athletes (2 males and 5 females)	Probiotic supplements after exercise	The increase in the number of *E. coli* in the athlete’s gastrointestinal tract, is likely to destroy the quantitative structure of the gastrointestinal flora, potentially ruining the stability and physiological function of the athlete’s gastrointestinal tract. Long-term oral probiotics for athletes can reduce inflammation in the body, reduce damage to the body during exercise, and effectively improve the gastrointestinal tract’s immune function.
Salleh, R.M. et al. [158]	2021	30 university badminton players aged from 19 to 22 years old	Probiotic supplements containing *Lactobacillus casei* Shirota	Supplementation of probiotics improved aerobic capacity in PG players by 5.9%, relieved anxiety and stress, but did not influence the speed, strength, leg power, and agility.
Hsu, Y.J. et al. [161]	2021	24 C57BL/6 J db/db male mice (20 weeks old)	*Bifidobacterium longum* OLP-01 (OLP-01)	Grip strength and exhaustive swimming time were significantly higher in mice that received exercise training combined with *Bifidobacterium longum* OLP-01 supplementation (EX OLP-01), comapred to mice in the other groups. EX OLP-01 showed effects on improved physical activity, lowered blood glucose, increased insulin sensitivity, decreased total body fat, and protection against liver injury, without any adverse effects.
Lee, M.C. et al. [162]	2021	40 six-week-old male ICR mice	Human-origin strain L. plantarum PL-02 supplementation	Four weeks of PL-02 supplementation could significantly improve grip and endurance exercise performance, and increase muscle mass, and hepatic and muscular glycogen storage. PL-02 could significantly decrease the level of fatigue indicators after exercise, such as lactate, BUN, ammonia, and CK.
Fu, S.K. et al. [163]	2021	8 recreational HM runners	4-week PS128 supplement administration	PS128 supplementation was related to an improvement in muscle damage, renal damage, and oxidative stress caused by HM, through microbiota modulation and related metabolites, but not in exercise capacity. PS128 may neutralize the ROS caused by exhaustive and prolonged exercise.
Valentino, T.R. et al. [164]	2021	42 four-month-old female C57BL/6J mice	Antibiotic cocktail treatment, consisting of 100 μg/mL each of metronidazole, neomycin, and ampicillin, and 50 μg/mL each of vancomycin and streptomycin	Gut microbiome dysbiosis caused by antibiotics treatment impaired the ability of skeletal muscle to adapt to exercise training. Dysbiosis caused blunted hypertrophy in both the soleus and plantaris muscles following PoWeR training, and was associated with a loss of PoWeR-induced myonuclei accretion in the plantaris muscle.
Quero, C.D. et al. [165]	2021	13 professional soccer players, 14 sedentary students	Synbiotic Gasteel Plus^®^ supplementation	Synbiotic nutritional supplements can improve anxiety, stress, and sleep quality, particularly in sportspeople, which appears to be associated with an improved immuno-neuroendocrine response involving IL-1β, CRH, and dopamine.

Abbreviations: BUN: blood urea nitrogen; CK: creatine kinase; ICR: Institute of Cancer Research; HCD: high-carbohydrate diet; HM: half marathon; HPD: high-protein diet; PL-02: Lactobacillus plantarum; PoWeR: progressive weighted wheel running; PS128: Lactobacillus plantarum PS128.

## Data Availability

The authors confirm that the data supporting the findings of this study are available within the article and/or its Appendix A.

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
