# Peer review of "A Bibliometric Analysis on the Research Trend of Exercise and the Gut Microbiome"

_microorganisms, 2023, doi:10.3390/microorganisms11040903_

Round 1

Reviewer 1 Report

Overview

The authors aimed to comprehensively capture the current status and trends in the gut microbiome and exercise. This study provides an objective assessment of the contributions of academic groups and individual researchers and reveals current research trends and hotspots, which will provide a basis for future research.

Bibliometric analysis is a popular and rigorous method for exploring and analyzing large volumes of scientific data.

The authors presented the bibliometric analysis adequately and consistently with the best practice guidelines.

Below are my comments. I have indicated with an asterisk the comments that contain revisions to be made.

Specific comments

*The title is too long and repetitive. I would entitle the manuscript "A bibliometric analysis on the research trend of exercise and the gut microbiome".

*Replace keywords with other words other than the title. For example, remove "exercise; gut microbiome; bibliometric analysis" and use new keywords that can optimize the search of the published manuscript through the search engine.

The aims and scope of the study are clear.

The scope of the study is large enough to warrant the use of bibliometric analysis.

The chosen bibliometric analysis techniques are appropriate to meet the aims and scope of the study.

The search terms have exemplified the scope of the study.

The coverage of the database was not adequate for the study because only WOSCC was used. However, the authors have stated this limitation in discussions.

*Readers cannot easily understand the bibliometric summary because figures (3-7) are not legible.

The writing aligns with the bibliometric summary presented.

The writing explains the peculiarities and implications of the bibliometric summary.

The writing aligns with the target outlet for publication.

Author Response

Thanks to the reviewer’s insightful suggestion. We have revised the manuscript title and replaced the keywords. We have replaced figures 3-7 with more legible figures.

Reviewer 2 Report

            The goal to incorporate physical activity and exercise is already established for a large diversity of human health conditions including metabolic, cardiovascular, neuromuscular, and many others.   The authors in the abstract state that to use exercise THROUGH intestinal microbiome intervention is not  without merit, but this concept is not presented to any degree in the data analysis and discussion by the authors.  What one reads in the text is data mining among a selection of articles that are loosely related.  The strength of the review is in the well designed and clearly presented study design.  The interpretation of the data is too strong and not suited for a Microbiology journal.   As the authors state, this is an informatics/communication type study, not one on either exercise nor the microbiome.  Thus my points are largely  devoted to making this clearer so that the authors could submit to a journal where possibly it would reach the intended audience.  IF they truly wish for health investigation of exercise and the microbiome, then much real scientific data from cited publications would be needed for all topics. 

            The data mining is largely observational without reference to concepts which the authors wish to advance, therefore this review is of limited importance and impact.  As one instance publications are listed from country of origin or institution of origin of research or senior authors.   This is not without some importance, but the authors do not comment on this.  They present statistics and graphical presentation, but do not delve into why the data is important. For this reviewer, one impacting factor this has is to bias publications towards one type of research and certain beliefs.   This can be substantiated or refuted by analysis of time frame and type and concepts of research being done and analyzing outcomes and data of research.   What will not be seen is that the use of exercise through changes in the intestinal microbiome may mediate a part of its effects.

            Perhaps if more meaningful parameters of the publications were used this bibliometric analysis would have more impact.   If instead of using institutions, authors, and keywords chosen by the authors, if bacterial types, bacterial metabolites, or newer technologies were used then the research in an area might be advanced.  One could over time track meaningful parameters from publications rather than totals.   Trends in specific and parameters determined by a number of published studies would enhance the quality and impact of publications.  This type of analysis is not included, the parameters impact and how they impact research evolution is not made clear. 

            The microbiological aspect of this review is very weak and does not present meaningful data even within the context of making future studies of exercise induced biome changes..  How different modes of exercise under differing conditions eg diets or background health factors, influence the results of the data is not presented in depth and with meaning.  Table 5 which is the 10 most cited publications demonstrates this point clearly.  Doing a Google Scholar search ‘exercise and intestinal microbiome’ all of these are found and many provide animal models, there are also human studies, on changes in the intestinal microbiome and potential  connections to health parameter changes by bacterial structures or secreted substances.  This meaningful data is what is neglected by this review rather than analyzing the sources of publication.    

            The review as it stands has little data on exercise modes, relation to how the intestinal microbiome might change, and how this might have health implications.   The authors state that their focus is a bibliometric analysis (as in lines 62-68) and useful for evaluation of research and focusing future research on needed topics.  This goal to this reviewer requires analysis of the content and goals of the published original research articles and the reviews in this field.   A superficial observational analysis as is presented without deeper scientific analysis on content, hypothesis, and speculation of discussion of most impactful publications in this field cannot achieve its goal.  Notably there is no discussion of the gut brain axis for the brain muscle axis of coordination of function. What this review does much better is an analysis of publications, authors and institutions, but there is more to be gleaned from studying this literature.  Trends in research in this area could be deciphered.  Implications of publications by one group on the field can also be deciphered.  This type of review would be appropriate when presented as analysis of how publications influence a discipline, in this case exercise and intestinal microbiome in health modulation.  It would be necessary to preface this that the review does not focus on exercise or the hiome and this cannot be assessed by such a review.   This review is as the authors state BIBLIOMETRIC and not focused on the science, therefore an informational and modes and impact of publication type publication is better suited to this review.  The authors use the term in the abstract ‘research topic evolution’ which is an interesting term but what it means and how this review impact the ‘evolution’ is not clear from the depth of analysis.  In addition there is the mixed message of improving and expanding the use of exercise and its possible mechanism of biome regulation to improve some facet of health.  However, how their data findings demonstrate this for exercise and microbiome alterations however is not clear. 

            If the authors had provided specific examples of how their bibliometric analysis was used to improve studies and practice in this field, it would be more apparent.  If they could demonstrate how a measured parameter in some basic animal model or human clinical study was used  and the data employed to make subsequent studies more impactful or solely more informative, this would strengthen the presentation greatly.  However, the depth of analysis of institutions and authors and  key words is too superficial and they have not demonstrated this goal of the study well.  There can be value in bibliometric analysis of any research area, this is not to be questioned.    How the specifics of bibliometric analysis are performed vary widely however all should have scientific depth and meaningful parameters for analysis. 

            It should also be pointed out that the English language used is unusual and may not clearly convey the intention of the authors.   The first example is the first word of the title, ‘spreading’.  If this would refer to the acceptance AND USE of a scientific or clinical concept or practice, this is not clear from this term.    The last sentence of the abstract is also very grandiose, broad, and unclear and demonstrates a non-conservative view of the impact of their analysis.   What do the terms ‘main trend’ or ‘hotspots in the future mean’?   These are strong statements and of a topic that is not the focus of the literature analysis rather than scientific content focused review. 

Author Response

Thanks to the reviewer’s insightful suggestion. We have completed the major revision of the manuscript according to the comments. We have revised the section “4.2. Hotspots and frontiers” in discussion. We analyzed the results of keyword analysis based on the latest research progress, and added the summary of recent advances in the field of exercise and gut microbiome. Table 6 has been added to summarize the recent cross-sectional studies in the field of gut microbiome and exercise. Table 7 has been added to summarize the recent longitudinal studies investigating the effect of exercise on gut microbiome. These two tables present the influence of exercise on bacterial types, bacterial metabolites, and potential connections to health parameter changes by bacterial structures or metabolites. We have also discussed the impact of different modes of exercise under different conditions (diets or background health factors) on gut microbiome and health states in this section. In addition, we focused on the impacts and advances of gut microbiome on exercise in Table 9. Besides, we have added the discussion of the recent progress in gut-brain axis and gut-brain-muscle axis.

Now we have completed the major revisions of content of the manuscript. We will apply the extensive English editing servise to further improve the quality of language in the manuscript.

Reviewer 3 Report

Reviewed article is very interesting as well as importent. It contains a bibliometric analysis on the research that describe the dependence between exercise and the gut microbiome. There is observed rapid increased number of publications which describe the influence of different factors, including exercise, on the composition of human gut microbiome, and on the role of gut microbiota in human health and diseases. Therefore, publication by Ruiyi Deng et al. is so recent. Reviewed manuscript needs few, however not important, corrections in text. For example, in lines 502, 505, and few other is: "Quiroga R et al. found...", "Huang JH et al. ...." respectively. It should be: "Quiroga et al. [90] found...", "Huang et al. [91]. This "error" is present few times in text. 

Author Response

Thanks to the reviewer’s insightful suggestion. We have corrected and unified format in the manuscript according to the suggestion.

Round 2

Reviewer 2 Report

The revision of this manuscript is a modest step towards making a greater impact with these studies.   The presentation is still largely observational and categorizes parameters on publications mostly at a more superficial level than on substantiative meaningful parameters.   The English revision helps greatly to make the points clearer, but there has been modest change to trying to make this bibliometric approach have a greater impact on the publications and therefore the nature and substance of studies in this field more meaningful and impactful.   Its  impact could be greater and future studies of this bibliometric type should aim to have more substantiative meaningful parameters as their focus and have sections devoted to why their studies will make publications and research more meaningful in any intellectual discipline.